# Allocation algorithms for multicore partitioned mixed-criticality real-time systems

Luis Ortiz, Ana Guasque, Patricia Balbastre and José Simó

Instituto de Automática e Informática Industrial (ai2), Universitat Politècnica de València, Valencia, Spain

## ABSTRACT

Multicore systems introduced a performance increase over previous monocore systems. As these systems are increasingly finding application in critical domains, it arises a necessity to develop novel methodologies for their efficient resource allocation. In addition, it is particularly important to consider the criticality of applications when scheduling such systems. In multicore systems, scheduling also includes the allocation of tasks to cores. In architectures based on spatial and temporal partitioning, it is also necessary to allocate partitions. Consideration of all these variables when scheduling a critical multicore partitioned system is a major challenge. In this article, a hypervisor partitioned framework for mixed-criticality systems is proposed. In this sense, the allocation process has been split in two different parts. The initial phase will allocate tasks to partitions according to the criticality of the system. This is achieved through the implementation of a Mixed-Integer Linear Programming (MILP) algorithm. The second phase involves the allocation of tasks to cores, employing both, an additional MILP algorithm and a modified worst fit decrease utilisation approach. Experimental results show that the combination of both strategies leads to feasible scheduling and, in addition, to a reduction of the overhead introduced by the hypervisor.

## INTRODUCTION

The use of multicore processors has been widely spreading across embedded systems, leveraging their processing capabilities to run applications on a single platform. The main weakness of these systems lays on the possible contention of resources, that are not found on monocore systems. This contention leads to alterations in the temporal behaviour of the task (*Dasari et al., 2013*).

In hard real-time systems these new processors are also getting introduced, but are impaired by these nondeterminisms. Specially since if a hard real-time task misses any temporal constraint in high criticality applications, may be disastrous. *Burns & Davis (2017)* defines criticality as the degree of assurance required to safeguard a system component against failure.

Partitioned systems in multicore processors, permit the division of the available cores and resources into separate partitions. Partitioned systems serve as a safeguard mechanism for temporal and spatial partitioning (TSP), wherein applications of different criticality can



Corresponding author
Ana Guasque, anguaor@ai2.upv.es

run isolated from each other within the same system. This isolation serves as a measure to avert the propagation of errors within the system.

In some cases, these systems might need to get certified by an external organisation, to prove that it behaves as expected. Commercial-off-the-shelf (COTS) components such as this new multicore processors offer advantages such as cost-effectiveness and faster time-to-market, their use in safety-critical applications raises concerns about reliability and compliance with safety standards, *Ye & Kelly (2004)* proposes a method for evaluating the feasibility of the use of COTS in the development of safety-critical systems. Conformance to a safety standard might be a requirement to get this certification, such as IEC 61508 for electronic systems, IEC 880 for nuclear power plants, DO-178B for airborne civil avionics, EN 50128 in European railways, ECSS for European space, *etc*. To mitigate the risk of cascading faults and to avoid the need of a system re-certification if it has been slightly modified, the employment of TSP is mandatory. TSP allows modification of systems already certified, where a change in a partition results in the re-certification of just this partition instead of the whole system.

Virtual machine technology can be considered the most secure and efficient way to achieve TSP. A hypervisor or virtual machine monitor (VMM) is a layer of software (or a mixture of hardware and software layer) that runs several partitions in a single computer. A great disadvantage of hypervisors is that they introduce an overhead (*Masmano et al., 2009*). In general, the overhead can be modelled with two components: the effect of the clock interrupt and the partition context switch (PCS). PCSs are defined as instances where a currently executing task is interrupted to allow a higher-priority task to run. This phenomenon occurs when a partition executing a lower-priority task is preempted by the activation of a task with greater priority, thereby ensuring that more critical processes are given immediate access to CPU resources.

The aim of this research is to optimize scheduling outcomes in the context of partition context switching, with particular emphasis on the criticality of tasks. The work is focus on critical systems, therefore scheduling will be static and migration between cores it is not allowed. This is due to the fact that it is a requirement to acquire certification. The scheduling on multicore partitioned platforms, is divided in two steps. Firstly, the tasks will be allocated into partitions and into cores, this will be the core of the article. Secondly the proper scheduling will be performed taking into account the data from the previous step, the scheduler, Earliest Deadline First (EDF) (*Liu & Layland, 1973*) slightly modified is going to be used. EDF will generate the static plan (*Hanen & Munier, 1993*) indicating where and when each task should be executed. One of the main advantages of the offline scheduling is the low execution cost at run time.

## Contribution

The main contributions presented in this article, are: a new method for allocating tasks to partitions, considering the criticality of these. This allocator is based on the MILP optimisation technique as well as one of the partitions to cores allocators explained in this article. Also, another allocator of partitions to cores is presented, this one is based on traditional bin packing techniques. Proposing both allocators allowed us to acquire

comparable data, since commonly used allocators does not take into account partitions. Therefore, those allocators would provide completely different results. Our goal is to generate allocations of tasks into cores in accordance with the systems safety requirements, that are schedulable and that provide a reduction in the partition context switches.

The rest of the article is organised as follows: The following section presents the related works found in literature. In the "Related works" the task model to be used and the problem are defined; the new methods proposed are presented in "Task Model and Problem Definition"; in the "Methods" an example is studied; the "Example" discusses the results; and finally the conclusions are provided in the "Conclusions".

## RELATED WORKS

Considerable research has been devoted to scheduling in real-time multicore systems, with one of the pivotal surveys in this domain being referenced as *Davis & Burns (2011)*. Within multicore scheduling, there are primarily two branches distinguished by the level of criticality demanded: partitioned and global scheduling. Given our focus on hard real-time applications, we will henceforth adopt partitioned scheduling. Partitioned systems (*e.g.*, those based on ARINC-653) are those in which migration of tasks between cores is not allowed. In these systems, once an allocation of tasks to cores has been achieved, is possible to use uniprocessor scheduling algorithms in most of the multiprocessor systems. This approach to multicore scheduling entails two key phases: task-to-core allocation and task scheduling for each core.

Bin packing algorithms are designed to address the NP-hard problem (*Johnson, 1973*; *Garey & Johnson, 1979*) associated with the allocation of tasks to distinct cores based on their utilisation, with the goal of optimising computational efficiency. Two of the most commonly used algorithms for solving this problem are First Fit (FF) and Worst Fit (WF) (*Coffman, Garey & Johnson, 1996*).

- The First Fit (FF) algorithm is a task allocation approach in which each task is allocated to the first core where it fits, without surpassing the utilisation threshold. In instances where allocation is not feasible on the current core, a new core may be utilised. This approach leads to an uneven distribution of the workload across the cores.
- The Worst Fit (WF) algorithm. Takes each task and allocates it into the core that leaves more remaining utilisation, opening more cores than FF. This approach leads to an even distribution of the workload.

The primary benefit of using a partitioning strategy in multicore scheduling is that once tasks are allocated to processors, numerous real-time scheduling techniques and analyses developed for monocore systems can be readily applied.

In a real-time and embedded system, the concurrent execution of processes with varying levels of criticality within the same platform is termed as a mixed criticality system (MCS) (*Burns & Davis, 2017*). This represents a significant and current trend, particularly evident in domains such as avionics and space exploration. Within these systems, applications with higher criticality incur greater costs for design and verification. However, mixed criticality systems offer several advantages, such cost-effectiveness, reduced space requirements,

lower weight, reduced power consumption and thus minimised heat generation. For instance, consider an aircraft where the in-flight information system holds lower criticality compared to the flight control systems, yet both operate within the same "mixed criticality" framework.

Regarding to the task scheduling of each core in MCS, the model typically used is the Vestal model. *Vestal (2008)* defined a task $\tau_i$ as an implementation of different functions and was defined, among others, by its criticality level and its worst case execution time (WCET). The WCET is denoted as a vector of values of WCET, in which each value corresponds to a criticality level. Typically, two levels of criticality are taken into account, namely high (HI) and low (LO), denoted as $C_i = (C_i^{LO}, C_i^{HI})$, with $C_i^{HI}$ being more conservative than $C_i^{LO}$. During runtime, the system initiates execution in the LO operation mode and must meet deadlines for both HI and LO tasks. If any HI task exceeds its $C_i^{LO}$ allocation, the system will transition to the HI operation mode, and LO tasks will be forfeited to ensure the deadlines of HI tasks are met.

This model relies on multiple execution times, and the underlying assumption is a topic of debate due to its impact on the practical applicability of research findings (*Ernst & Natale, 2016*). The measurement of two values of $C_i$ necessitates two distinct processes: a simpler estimation process for $C_i^{LO}$ and a more rigorous one for $C_i^{HI}$, incorporating more pessimistic assumptions. The certification process, being a critical aspect, poses challenges when dealing with the acceptance of two different $C_i$ values through separate processes. For instance, if certification authorities (CAs) approve $C_i^{LO}$ for a high-critical task, there may be no need to introduce additional pessimism, and vice versa. However, if CAs endorse $C_i^{HI}$ for a high-critical task, they are unlikely to accept a lower value like $C_i^{LO}$ that could compromise the overall process.

The following works are based on Vestal model and apply bin packing techniques to solve the allocation problem: *Baruah et al. (2014)* solves the partition-to-cores allocation following the next criteria: first, the algorithm allocates HI tasks in cores using First-Fit (FF) algorithm. Then, it repeats the operation with LO tasks. It has to be checked that system utilisation per mode do not exceed 3/4. After mapping, Earliest Deadline First with Virtual Deadlines (EDF-VD) (*Baruah et al., 2012*) scheduling is applied to each core.

*Gu et al. (2014)* first uses Worst-Fit (WF) packing strategy to allocate HI tasks and tune the virtual deadlines to HI tasks in each processor. Then it uses FF to allocate LO tasks.

The main difference between these works and our work is that this work does not follow Vestal model due to it is "hardly acceptable in practice" (*Ernst & Natale, 2016*) and the difficulty in the certification process. In this sense, our model considers different criticality levels following safety standards such as DO-178B (*Rierson, 2017*) and considers a single WCET for each task. Moreover, they use bin-packing algorithms to allocate tasks to cores, in which tasks criticalities are not considered. *Peng, Shin & Abdelzaher (1997)* solves the problem of allocating communicating periodic tasks to heterogeneous processing nodes in a distributed real-time system. In this work, tasks are modeled with task graphs to cope with precedence constraints, they work with heterogeneous distributed systems and propose an optimal branch and bound approach for scheduling periodic task graphs, which is out of the scope of this article and does not fit exactly to the characteristics of our

partitioned system with different criticalities. In this same line, *Senoussaoui et al. (2020)* considers multiprocessor systems with identical cores, where each core has its own scheduler and queue. So they work as if it were a single processor, that is, without interference. We consider the problem of interference due to hardware shared resources, which is one of the main challenges of multicore systems, and also consider tasks with different criticalities.

In addition to bin-packing allocation algorithms, there are non-conventional techniques to solve the multicore scheduling problem. For example, techniques such as Mixed Integer Linear Programming (MILP), which is an optimisation technique that defines constraints and an objective function and minimises/maximises different variables. Some works use MILP to customise the allocation and scheduling problem to applications. In most cases, task priority assignment will be defined as variables (*Zhu et al., 2013*; *Rivas et al., 2024*), with the worst-case response time (WCRT) considered either as constraints or an objective function. This setup aims to determine the optimal priority assignment to minimise response time. *Guasque et al. (2020)* proposes an ILP method that obtains a static schedule for periodic tasks and partitioned systems where temporal and spatial isolation is crucial. However, they are focused on monocore systems and the reduction of task context switches and not partitioning, as this article does. Other articles of the same authors (*Aceituno et al., 2021*, *2022*) extend the model to multicore systems, but they do not consider partitioning models.

Despite the integration of advanced techniques into MILP solvers like CPLEX (*Cplex & IBM ILOG, 2009*) or Gurobi (*Gurobi Optimization, LLC, 2023*), these solvers still struggle to handle large-scale cases (*Baruah, 2020*). *Zhao & Zeng (2017)* introduced several problem-specific insights that serve as the basis for a more efficient framework. For instance, while the calculation of WCRT is known to be NP-hard, it often proves to be reasonably efficient in practice (*Davis, Zabos & Burns, 2008*). However, conventional ILP solvers fail to exploit this observation. The article also presents an efficient algorithm for determining the maximum virtual deadline that renders the task system unschedulable. By integrating these insights with a classic ILP solver, the framework achieves significant efficiency gains.

*Davare et al. (2006)* considers a number of MILP approaches for solving the task allocation and scheduling problem for a Xilinx FPGA platform. In this work, the MILP formulation is customised for a multimedia case study which is out of the scope of this article as we consider critical systems and moreover is applied to a specific platform, and we consider a general model.

In *Mangeruca et al. (2007)*, task precedence relations are taken into account. This study utilises relaxed ILP techniques to derive an optimal assignment of priority/deadline for preemptive dynamic priority scheduling while accommodating precedence constraints. Additionally, *Lisper & Mellgren (2001)* addresses the problem of response time calculation and priority assignment using ILP, where the ceiling of the response time equation is redefined as an ILP problem. Furthermore, an enhanced real-time schedulability test is proposed in *Zeng & Di Natale (2013)*, allowing for a more precise and efficient delineation of the feasible region with fewer binary variables. These works consider monocore systems

and our work is extended to multicore systems. Furthermore, our objective extends beyond merely identifying a feasible schedule; we aim to discover the improved schedule considering various objectives.

## TASK MODEL AND PROBLEM DEFINITION

A multicore system is supposed with $m$ cores $(M_0, M_1, M_2, \ldots, M_{m-1})$, where a task set $\tau$ of $n$ periodic and independent tasks should be allocated. Each task $\tau_i$ is represented by the tuple:

$$\tau_i = (C_i, D_i, T_i, I_i, Cr_i) \tag{1}$$

where $C_i$ is the WCET, $D_i$ is the relative deadline, $T_i$ is the period, $I_i$ is the interference and $Cr_i$ is the level of criticality of the task.

The term $I_i$ is the interference, defined as in *Aceituno et al. (2021)*. It is the time the task takes to access shared hardware resources. A typical case is the operation of reading and writing in memory.

The term $Cr_i$ is meant to be the level of criticality according to a safety standard as mentioned in the introduction section, in our case we will use three levels based on DO-178B, that are A, B and C, being A top catastrophic, B mayor impact and C non-critical. Note that although DO-178B has five criticality levels, we use only three for the sake of simplicity.

The utilisation of a task $\tau_i$, is expressed as the quotient of the computation time divided by the period, $U_i = \frac{C_i}{T_i}$. Since the $I_i$ is modelled, the real utilisation would be: $U_i' = U_i + U_i^{\text{int}}$, that includes the utilisation due to the interference. The utilisation of a core $M_k$ will be the sum of the utilisation of all the tasks assigned to this core, $U_{\tau_{M_k}} = \sum_{\tau_i \in M_k} U_i$. The real utilisation of a core $M_k$ will be the sum of the real utilisation of all the tasks assigned to this core, $U'_{\tau_{M_k}} = \sum_{\tau_i \in M_k} U_i'$. Therefore the total utilisation of the system is the addition of all the utilisation of all the cores, $U_\tau = \sum_{\forall k} U_{\tau_{M_k}}$ and the real utilisation of the system is $U'_\tau = \sum_{\forall k} U'_{\tau_{M_k}}$.

Since we are working with partitioned systems, prior to the allocation of tasks to cores, tasks are allocated to partitions. Partitions are defined as $P_j$. A partition aims to provide spatial and temporal isolation from the rest of the system increasing therefore the security of it. Spatial isolation refers to the allocation of dedicated memory to each partition, ensuring that no other partition or subsystem can access it. On the other hand, temporal isolation involves assigning a specific time window for the execution of each partition, preventing it from exceeding its allocated time frame. A system may consist of as many partitions as needed, denoted as $P_0, P_1, P_2, \ldots, P_j$. Hypervisors are software layers capable of virtualising the hardware in order to be able to create different virtual machines. They are in charge of the partition management, ensuring the temporal and spatial partitioning (TSP).

As a consequence of the TSP, the phenomenon of partition context switches (PCS) appear. PCS are characterised by the alternation of the execution of partitions, driven by system requirements.

Therefore the primary objective of this study is to introduce an innovative methodology for the allocation of tasks to partitions and subsequently to cores, focusing on minimising

the frequency of partition context switches. This way, it is intended to allocate tasks of the same partition in the same core.

# METHODS

This section presents the methodology to solve the allocation of tasks to cores problem, considering the level of criticality of the tasks in order to form the appropriate partitions. The first step consists of allocating the tasks to partitions. The second step consists of applying any task allocation to cores algorithm for multicore systems. Different allocation algorithms are proposed and compared in this step.

## Tasks to partitions allocation

In this allocation problem, first, tasks are grouped into partitions, taking into account that tasks of different criticality may not belong to the same partition. In this section, we will first present a MILP model that provides the optimal allocation of tasks to partitions, considering the criticality of the tasks. Table 1 introduces the different indices, parameters, and variables used in the model.

According to the problem statement, the objective function is defined in Eq. (2), which is maximising the number of tasks per partition maintaining the criticality condition.

$$\max \quad Obj = \sum_{\forall i,j} a_{ij} \cdot p_j \tag{2}$$

s.t:

$$\sum_i a_{ij} U_i = U_j \quad \forall j$$
$$U_j \leq 1, \qquad \forall j \tag{3}$$

$$\sum_{\forall j} a_{ij} = 1 \quad \forall i \tag{4}$$

$$a_{ij} + a_{iij} \leq 1 \quad \forall i, ii \in \tau, \forall j \text{ if } Cr_{\tau_i} \neq Cr_{\tau_{ii}} \text{ and } i > ii \tag{5}$$

$$\sum_{\forall i} a_{ij} = p_j \quad \forall j \tag{6}$$

$$a_{ij} \in \{0, 1\} \tag{7}$$

$$U_j, p_j \geq 0. \tag{8}$$

Equation (3) calculates the partition utilisation, which is the sum of the utilisations of the tasks that belong to that partition. This utilisation must be less than or equal to 1. Each task must belong to a single partition (Eq. (4)). Equation (5) ensures that two tasks with different criticality are not grouped in the same partition. Equation (6) counts the number of tasks in each partition. Equations (7) and (8) represent the decision variable domains.

## Tasks to cores allocation

Once the tasks are grouped into partitions, the next step consists of allocating tasks to cores. First, let us present some considerations about allocation algorithms and then, we introduce our proposals.

**Table 1 Model notation of the MILP task-to-partition allocation problem.**

| Sets and indices | |
|---|---|
| $i$ | Tasks $\tau_i \in \{0, 1, 2, \ldots, n-1\}$ |
| $j$ | Partitions $P_j \in \{0, 1, 2, \ldots, n-1\}$ |
| **Parameters** | |
| $U_i$ | Utilisation of $\tau_i$ |
| $Cr_i$ | Criticality of $\tau_i$ |
| **Decision variables** | |
| $a_{ij}$ | 1 if $\tau_i$ is allocated to partition $P_j$ and 0 otherwise. |
| $U_j$ | Utilisation of $\tau_j$. |
| $p_j$ | Number of tasks allocated to partition $P_j$. |

### Bin-packing algorithms

As mentioned in the related works section, bin packing algorithms only consider the utilisation of the tasks, what makes them sensitive to the order in which tasks are sorted. Therefore, new variants of these algorithms can be produced based on that, *i.e.*, decreasing or increasing utilisation. The allocating heuristics known as First Fit Decreasing Utilisation (FFDU) and Worst Fit Decreasing Utilisation (WFDU) (*Oh & Son, 1995*; *Coffman, Garey & Johnson, 1996*) arrange tasks, $\tau_i$, in a descending order based on their utilisation $U_i$. Therefore, these heuristics prioritise tasks with higher utilisation values.

### Algorithm based on WFDU, WFDU'

As stated before Worst Fit Decreasing Utilisation, arrange tasks according to their utilisation on the system. In the course of our investigations, we opted to implement a slightly adapted variant called WFDU'. In this modified version, the task utilisation's are aggregated within each partition, based on the partition assignation from the first MILP algorithm, and subsequently, the partitions are allocated to the cores based on their total utilisation. Such implementation provides us with another allocation method comparable to the MILP presented below.

### Integer Linear Programming based algorithm

Once the tasks are grouped into partitions, the next step consists of allocating tasks to cores, considering the partition to which the task belongs. Table 2 introduces the different indices, parameters, and variables used in the model. According to the problem statement, the objective function is defined in Eq. (9), which is minimising the number of partitions per core.

$$\min \quad \text{Obj} = \sum_{\forall j, k} q_{jk} \tag{9}$$

s.t:

$$\sum_{\forall i} o_{ik} U_i = U_{\tau_{M_k}} \forall k \tag{10}$$

$$U_{\tau_{M_k}} \leq 1 \forall k$$

**Table 2 Model notation of the MILP task-to-core allocation problem.**

**Sets and indices**

| | |
|---|---|
| $i$ | Tasks $\tau_i \in \{0, 1, 2, ..., n-1\}$ |
| $k$ | Cores $M_k \in \{0, 1, 2, ..., m-1\}$ |
| $j$ | Partitions $P_j \in \{P_{\tau_0}, ..., P_{\tau_{n-1}}\}$ |

**Parameters**

| | |
|---|---|
| $U_i$ | Utilisation of $\tau_i$ |
| $P_{\tau_i}$ | Partition to which the task $\tau_i$ belongs |

**Decision variables**

| | |
|---|---|
| $o_{ik}$ | 1 if $\tau_i$ is allocated to core $M_k$ and 0 otherwise. |
| $U_{\tau_{M_k}}$ | Utilisation of core $M_k$. |
| $q_{jk}$ | 1 if tasks that belong to partition $P_j$ are allocated to core $M_k$ and 0 otherwise. |

$$\sum_{\forall k} o_{ik} = 1 \quad \forall i \tag{11}$$

$$q_{jk} = 1 \qquad \forall i, j, k \text{ if } o_{ik} = 1 \text{ and } j = P_{\tau_i} \tag{12}$$

$$o_{ik}, q_{jk} \in \{0, 1\} \tag{13}$$

$$U_{\tau_{M_k}} \geq 0. \tag{14}$$

Equation (10) calculates the core utilisation, which is the sum of the utilisations of the tasks that are allocated to that core. This utilisation must be less than or equal to 1. Each task must belong to a single core (Eq. (11)). Equation (12) builds an array that relates cores and partitions. If a task $i$ is allocated to a core $k$, *i.e.*, $o_{ik} = 1$, then the partition to which that task $i$ belongs will belong to that core. Equations (13) and (14) represent the decision variable domains.

## EXAMPLE

This section aims to provide with an example a simple comparison between both allocators. This example consist of a system with eight tasks that must be grouped in partitions and allocated to two cores, as the one defined in Table 3.

Due to their level of criticality and utilisations, tasks are grouped in three partitions as described in Fig. 1.

Once tasks are grouped into partitions considering the criticality, tasks must be allocated to cores.

### Tasks allocation with WFDU'

As explained before, the first step is to rearrange the table according to the utilisation of the tasks, as seen in Table 4.

Hence, the first task to be assigned would be T2, and it will be assigned to Core 1, followed by the rest of the tasks from the same partition. Subsequently, the ensuing task in the allocation sequence is T0, which is allocated to Core 0, along with the allocation of all

**Table 3 Example of a set of tasks to group in partitions and allocate to cores.**

|      | Criticality | Utilisation |
| ---- | ----------- | ----------- |
| T0   | A           | 0.45        |
| T1   | B           | 0.08        |
| T2   | C           | 0.467       |
| T3   | A           | 0.15        |
| T4   | C           | 0.032       |
| T5   | C           | 0.05        |
| T6   | C           | 0.0575      |
| T7   | C           | 0.035       |

Partition 0    Partition 1    Partition 2

$T_1$    $T_0$ $T_3$    $T_2$ $T_4$ $T_5$ $T_6$ $T_7$

**(a)** Representation of the partitions

|             | Tasks              | Level of criticality | Total utilisation |
| ----------- | ------------------ | -------------------- | ----------------- |
| Partition 0 | T1                 | B                    | 0.08              |
| Partition 1 | T0, T3             | A                    | 0.6               |
| Partition 2 | T2, T4, T5, T6, T7 | C                    | 0.6412            |

**(b)** Definition of partitions and utilisations.

**Figure 1 Partitions for the task set defined in Table 3.**

**Table 4 Table ordered by decreasing utilisation.**

|      | Criticality | Utilisation |
| ---- | ----------- | ----------- |
| T2   | C           | 0.467       |
| T0   | A           | 0.45        |
| T3   | A           | 0.15        |
| T1   | B           | 0.08        |
| T6   | C           | 0.0575      |
| T5   | C           | 0.05        |
| T7   | C           | 0.035       |
| T4   | C           | 0.032       |

tasks originating from the same partition. Finally, following the principles of Worst-Fit (WF), given the utilisation metrics of the cores, $U_{\tau_{M_0}} = 0.6415$ and $U_{\tau_{M_1}} = 0.6$, task T1 is directed to Core 0. Leading to the allocation presented in Fig. 2.
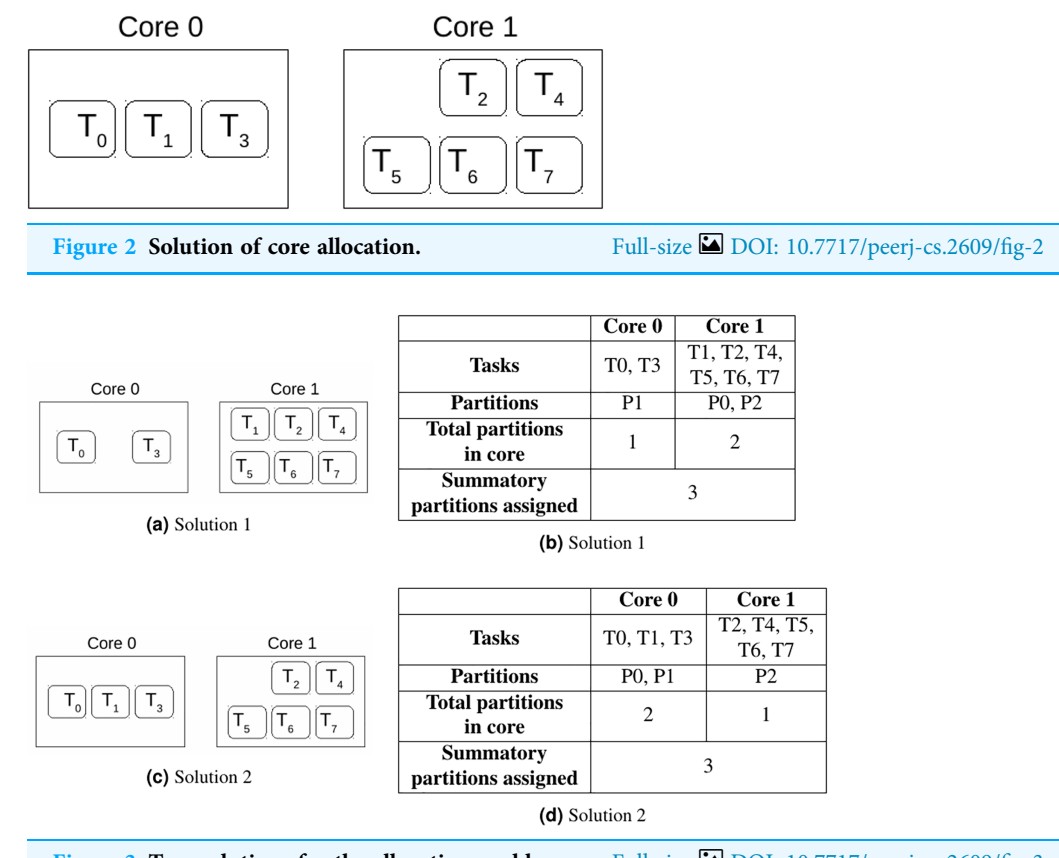

Figure 2 **Solution of core allocation.**       

(a) Solution 1

| | Core 0 | Core 1 |
|---|---|---|
| **Tasks** | T0, T3 | T1, T2, T4, T5, T6, T7 |
| **Partitions** | P1 | P0, P2 |
| **Total partitions in core** | 1 | 2 |
| **Summary partitions assigned** | 3 | |

(b) Solution 1

(c) Solution 2

| | Core 0 | Core 1 |
|---|---|---|
| **Tasks** | T0, T1, T3 | T2, T4, T5, T6, T7 |
| **Partitions** | P0, P1 | P2 |
| **Total partitions in core** | 2 | 1 |
| **Summary partitions assigned** | 3 | |

(d) Solution 2

Figure 3 **Two solutions for the allocation problem.** 

## Tasks allocation with MILP

As stated before, this problem minimises the number of different partitions allocated to cores, in order to minimise the number of partition context switches. For this purpose, the objective function in Eq. (9) minimises the sum of the $q$ array. However, there may be multiple solutions with the same objective value. As seen in Fig. 3, there are two possible solutions for the same allocation problem, and both of them provide the same objective value. The total partition context switches is equal to 3. Therefore, which is the best solution in terms of total partition context switches? The actual total number of context switches is determined once the scheduling phase is done. In order to select the solution that reduces them as much as possible, let us present the proposed methodology in Algorithm 1.

Once the task set $\tau$ has been grouped into partitions (lines 1–5), tasks must be allocated to cores. The algorithm (line 6) receives the task set, the number of available cores and the partition to which each task belongs. Then, the model is created and the variables and constraints are added (lines 7–8). The number of solutions (nSol) to be found is established (line 9) and then the search starts (line 10). If the solution is reached, the task set is evaluated with the resulting allocation of task to cores and the scheduling plan is obtained. Then, the actual number of partition context switches is measured and the schedulable solution with the least number of context switches is stored.

---

**Algorithm 1  MILP algorithms.**

1:  **procedure** TASKS INTO PARTITIONS ALGORITHM: T

2:      Create the MILP model. Add variables.

3:      Apply Eqs. (2)–(8) to group tasks into partitions considering the level of criticality

4:      Set objective and optimise $\rightarrow \{P_{\tau_0}, ..., P_{\tau_{n-1}}\}$;

5:      **return** $\{P_{\tau_0}, ..., P_{\tau_{n-1}}\}$;

6:  **procedure** TASKS TO CORES ALGORITHM: $(\tau, m, \{P_{\tau_0}, ..., P_{\tau_{n-1}}\})$

7:      Create the MILP model. Add variables.

8:      Apply Eqs. (9)–(14) to allocate tasks $\tau$ to $m$ cores.

9:      Select mode for exploring the MILP search tree and find the $nSol$ best solutions (with no relative gap for stored solutions).

10:     Set objective and optimise $\rightarrow$ Allocation $O_{\tau m}$.

11:     **if** model is infeasible **then**

12:         Exit;

13:     **else**

14:         nSol=model.SolCount                                    ▷Get the number of solutions

15:         **for** $s_i \in$ nSol **do**

16:             Obtain scheduling plan of $\tau$ allocated to $m$ cores, $O_{\tau m}$

17:             Calculate the number of context switches from the scheduling plan, $nCC_{s_i}$

18:             **if** $nCC_{s_i} < minCC$ **then**

19:                 $minCC = nCC_{s_i}$;

20:         **return** Allocation $O_{\tau m}$ with $minCC$

21: **procedure** MAIN:

22:      INPUT: Task set with $n$ tasks characterised by (C,D,T,I,Cr)

23:      OUTPUT: Allocation of tasks into partitions and cores with minimisation of partition context switches

24:      $\{P_{\tau_0}, ..., P_{\tau_{n-1}}\}$ = TASKS INTO PARTITIONS ALGORITHM $(\tau)$.

25:      $O_{\tau m}$ = TASKS TO CORES ALGORITHM $(\tau, m, \{P_{\tau_0}, ..., P_{\tau_{n-1}}\})$.

---

# RESULTS

## Experimental conditions

Figure 4 illustrates the simulation environment utilized in our study. The simulation process is initiated with load generation, and since we are working on a hypervisor framework is followed by task allocation, which is conducted in two stages: initially to partitions and subsequently to the specific core designated for execution. Following task allocation, the task set undergoes scheduling and validation to ensure the accuracy and reliability of the results. Each of these processes is elaborated in detail in the subsequent sections.

**Load generation.** The load is produced through a synthetic task generator, with the allocation of tasks to cores influencing both the number of tasks in each set and the overall system utilisation. The distribution of utilisation among tasks follows the UUniFast

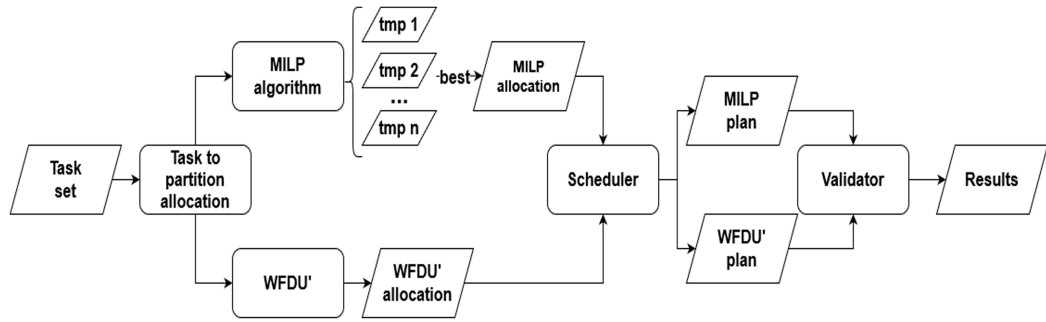

**Figure 4  Experimental simulation environment.**     

discard algorithm of *Davis & Burns (2009)*, based on the given total system utilisation and the number of tasks per set. Randomly generated periods fall within the range [20,1000], and computation times are derived from the system utilisation, while deadlines are constrained to be less than or equal to the corresponding periods. Also for the sake of simplicity of the task-set creation process, $Cr_i$ term will be assigned randomly.

Table 5 outlines the experimental parameters for the evaluation process.

The theoretical utilisation spans from 50% to 75% of the system's maximum load capacity. For instance, in a four-core system with a maximum load of 4, the initial utilisation for evaluation is set at 2.1 ($\approx$50%) and 3 (75%).

The percentage of broadcasting tasks is established at 25% of the total task count, except for scenarios 1–6 (two cores), where it is set at 50%. This adjustment accounts for the absence of interference when only one task is broadcasting. Each configuration of cores and utilisation is assessed with 10%, 20%, and 30% interference over the WCET. It is noteworthy that while not all tasks within a set share the same interference value, they all experience the same percentage of interference relative to the WCET.

**Allocation phase.** First, tasks are allocated to partitions following the algorithm described with Eqs. (2)–(8). Then, tasks are allocated to cores following WFDU and MILP algorithms described with Eqs. (9)–(14). Therefore, two possible allocations are generated.

**Scheduling phase.** Each task set moves to the scheduling step, where a scheduling plan is generated for each of the proposed allocations, *i.e.*, a MILP plan and a WFDU' plan.

The scheduling phase employs the contention-aware scheduling algorithm introduced in *Aceituno et al. (2021)*. This is a scheduling algorithm that considers the exact interference produced for each task. It is based on a priority-based algorithm, to select the task to run on each core. More details can be found in *Aceituno et al. (2021)*. Given that any priority-based algorithm can serve as the foundation for this approach, we have opted for EDF algorithm (*Liu & Layland, 1973*). The scheduler produces a temporal plan, detailing the scheduling of tasks for each core at each point in time. This plan undergoes validation to ensure that all temporal constraints are met.

The actual utilisation of the system is defined as $U'_\tau$ and is determined after the scheduling phase, when the actual interference is measured by comparing if different tasks on different cores activate at the same time. This utilisation is always greater or equal than

**Table 5 Definition of the experimental scenarios.**

| Number of cores | Utilisation | Number of tasks | Number of broadcasting tasks | Interference over WCET (%) | Scenario |
|---|---|---|---|---|---|
| 2 | 1.1 | 4 | 2 | 10 | 1 |
|  |  |  |  | 20 | 2 |
|  |  |  |  | 30 | 3 |
|  | 1.5 |  |  | 10 | 4 |
|  |  |  |  | 20 | 5 |
|  |  |  |  | 30 | 6 |
| 4 | 2.1 | 12 | 3 | 10 | 7 |
|  |  |  |  | 20 | 8 |
|  |  |  |  | 30 | 9 |
|  | 3 |  |  | 10 | 10 |
|  |  |  |  | 20 | 11 |
|  |  |  |  | 30 | 12 |
| 8 | 4.1 | 20 | 5 | 10 | 13 |
|  |  |  |  | 20 | 14 |
|  |  |  |  | 30 | 15 |
|  | 6 |  |  | 10 | 16 |
|  |  |  |  | 20 | 17 |
|  |  |  |  | 30 | 18 |
| 10 | 5.1 | 28 | 7 | 10 | 19 |
|  |  |  |  | 20 | 20 |
|  |  |  |  | 30 | 21 |
|  | 7.5 |  |  | 10 | 22 |
|  |  |  |  | 20 | 23 |
|  |  |  |  | 30 | 24 |

$U_\tau$, because it does not consider the interference. Therefore, the increased utilisation is measured as:

$$\alpha(\%) = \frac{U'_\tau - U_\tau}{U_\tau} \cdot 100 \tag{15}$$

that is, it is the ratio between the actual utilisation and the theoretical utilisation.

## Experimental results

This study leverages the Gurobi solver, which is accessible for students, faculty, and researchers at no expense, enabling them to engage in mathematical optimisation. We will be using Gurobi to solve the previously developed MILP algorithms. The experiments are conducted on a system equipped with an Intel Core i7 3.2 GHz processor and 32 GB RAM, utilising Gurobi 11.0.1 as the Mixed-Integer Linear Programming (MILP) solver.

For validating our results we have proposed three different metrics, schedulability, number of context switches and mean increased utilisation. All this metrics have been evaluated in a set of scenarios depicted on Table 5.

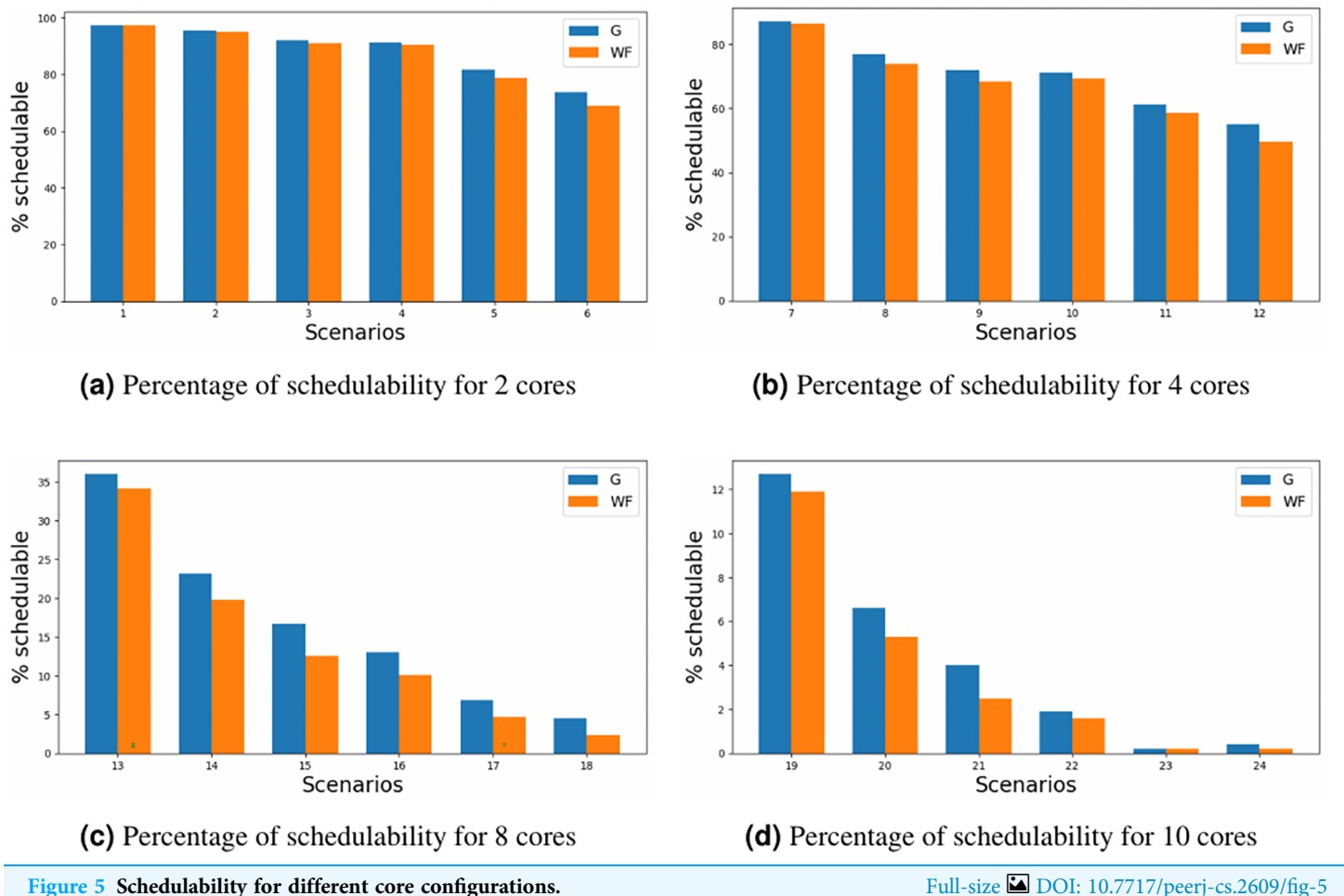

**Figure 5  Schedulability for different core configurations.**               

The results are depicted for schedulability in Fig. 5, where the y-axis represents the percentage of schedulability for each of the scenario' number (x-axis) defined in Table 5. Upon comparative analysis of the two allocators, it is evident that MILP allocator consistently achieves slight superior schedulability rates in contrast to the WFDU' allocator. This disparity becomes more pronounced as core or loads are increased. However, it is noteworthy an exception to this trend in the instance of scheduling 10 cores, wherein the challenge of scheduling such amount of cores, mitigates the observed performance advantage in heavily loaded systems. Although the WFDU' allocator should balance the load hence increasing the schedulability, due to the assignation of three types of criticality and the assignation to cores depending on the partition, there might be no load balancing effect compared to the use of WFDU.

The primary objective of this study was to minimise the number of partition context switches. Although the results presented in Fig. 6 may not immediately appear promising; where the y-axis represents the number of partition context switches for each of the scenario' number (x-axis) defined in Table 5; some considerations need to be made. Firstly, the comparison exclusively accounts for cases where both allocators generated a valid

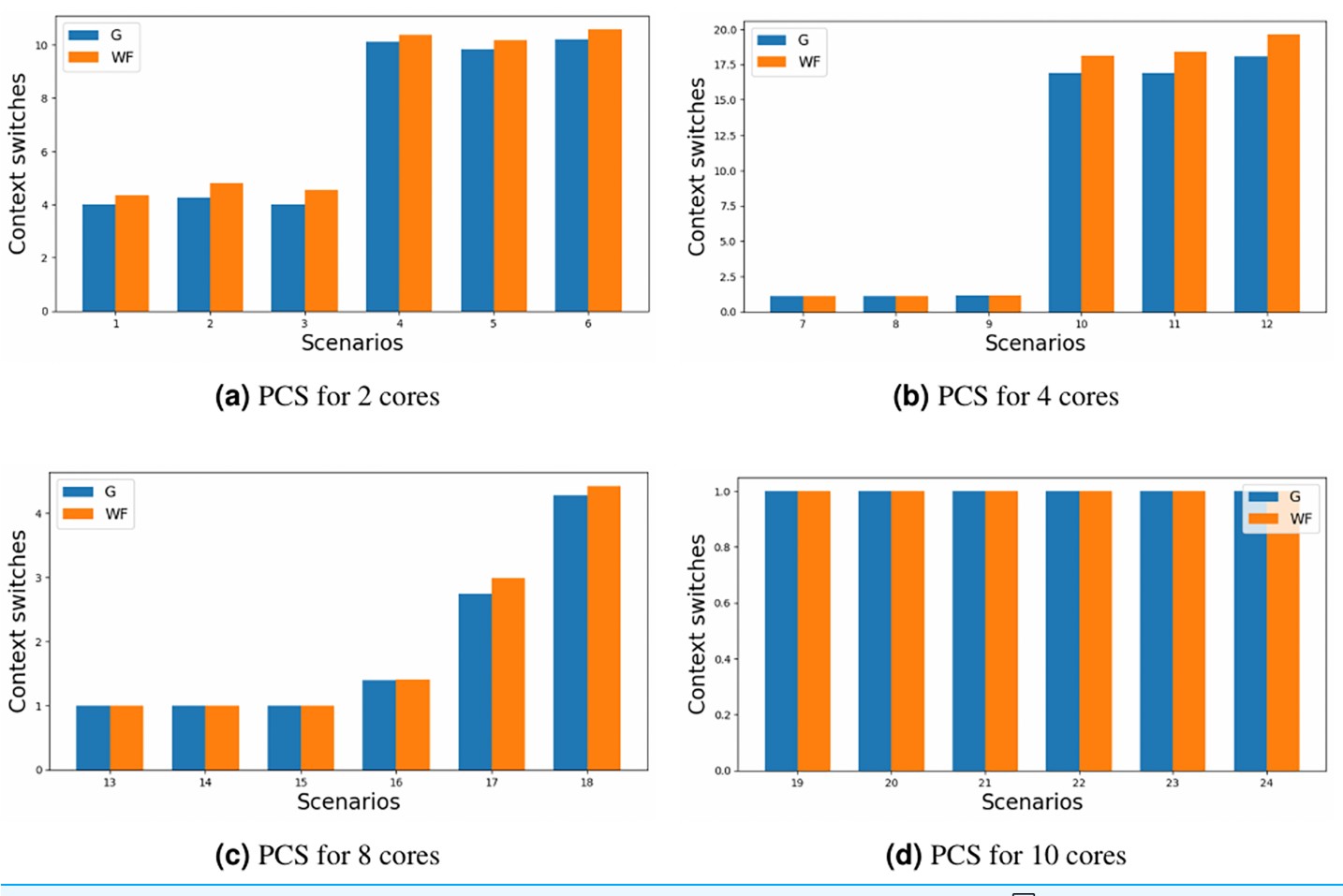

**Figure 6 Partition context switches for different core configurations.**

schedule. Secondly, the limited number of criticality levels (only 3) implies that partitions have the opportunity to exclusively occupy a core, particularly in scenarios characterised by lower core loads or a substantial number of cores. Consequently, attention should be directed towards configurations involving two and four cores, especially on the heavy loaded ones, where the cores might be outnumbered by the possible partitions.

The impact on the increase in utilisation should also be taken into account. MILP allocator introduces a lower amount of utilisation to the system, based on Eq. (15), in Fig. 7 it can be appreciated that the maximum increased utilisation is around 6% in scenario 3, where as with WFDU' is slightly higher. In this figure the y-axis represents the increased utilisation for each of the scenario' number (x-axis) defined in Table 5. This is due to the fact that WFDU' allocations tend to be more balanced in terms of load than MILP allocations. Then, as the load is shared among cores, there is more chance that tasks will interfere with each other. Therefore, the actual utilisation of the system will increase to a greater extent than MILP does.

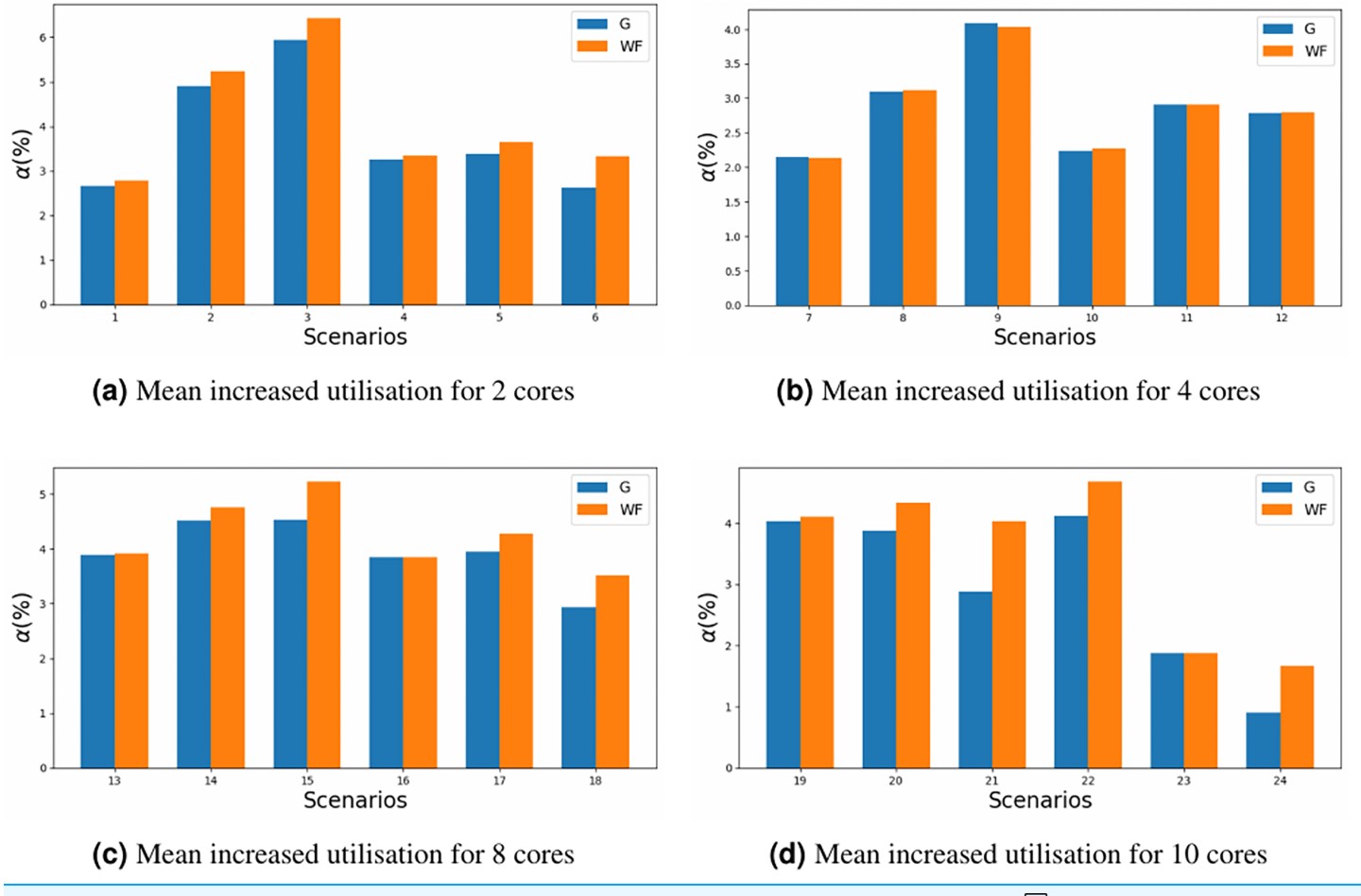

**Figure 7** Mean increased utilisation for different core configurations.

## Runtime and complexity of the algorithms

This article introduces three algorithms designed for task allocation across computing cores. In this section the runtime and complexity of each of this would be explained.

Algorithm 1 employs a Mixed-Integer Linear Programming (MILP) formulation, where the problem size scales quadratically with the number of tasks. The real complexity of the problem can not be accurately determined because of its strong dependence on the effectiveness of the pruning heuristic that Gurobi applies internally. On the other hand, an experimental analysis (*Till et al., 2003*), as shown in Fig. 8, reveals that increasing the number of cores leads to an increase in execution time in an abrupt manner with a noticeable peak when 10 cores are used. For the complexity and runtime experiments, the parameters also defined in Table 5 have been used.

Comparatively, the tasks-to-cores algorithm (described as a procedure in Algorithm 1–line 6), also based on MILP but featuring an iterative structure across a set of solutions, nSol, and its complexity is dominated by: nSol, the length of the scheduling plan (hyperperiod H), and the number of cores m, resulting in an approximate complexity of O

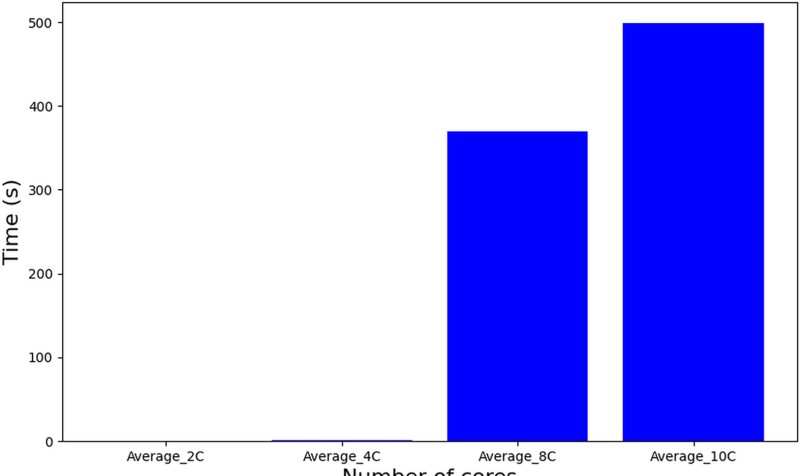

**Figure 8** **Runtime Algorithm 1.**

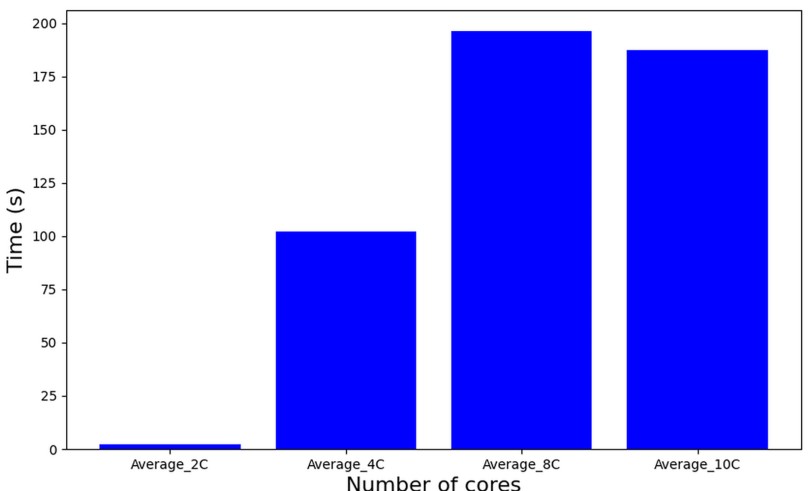

**Figure 9** **Runtime of tasks-to-cores algorithm.**

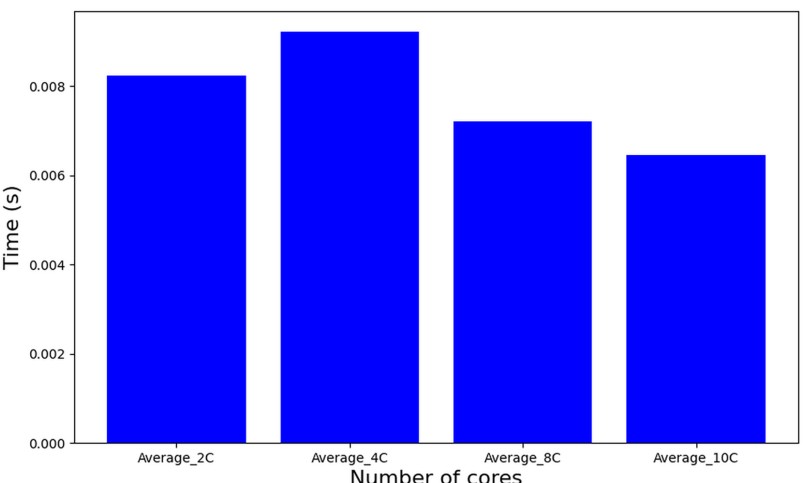

**Figure 10** **Runtime algorithm WFDU'.**

(nSol· H· m). This escalation in problem complexity manifests in substantially longer execution times, as depicted in Fig. 9. It is important to note that a significant increase in complexity occurs when using 10 cores or higher, which leads to a diminishing number of feasible solutions. Consequently, this results in a reduction in the computation time required for further processing.

In contrast to the MILP-based approaches, a heuristic method such as WFDU' offers an alternative allocation strategy. As depicted in Fig. 10, this heuristic demonstrates markedly reduced computational requirements. Specifically, it operates approximately 20,000 times faster than tasks-to-cores algorithm, underscoring its computational efficiency in comparison to exact MILP formulations.

## CONCLUSIONS

In this article, we have explored different techniques to allocate tasks of diverse criticality to cores in hard real-time systems. Our goal was the reduction of PCS, that we achieved by the combination of a two step allocator. We have proposed a first MILP technique that tries to group into partitions the tasks of the same level of criticality, what is the key part of this article since it produces major reduction. Secondly we proposed two methods for allocating partitions to cores, one based on MILP techniques and the other one a modification of WFDU, both delivering great performance in the reduction. The conclusion is that the most important factor in reducing partition context switches is to pack in the cores the minimal amount of different partitions. As possible expansions of the presented work, we could model the impact of the time of swapping partitions, also since we are working with a model that contains the interference, modifications on the maximal quantity of utilisation of a core can be proposed.

As future work, the use of MILP techniques to reduce not only context switches but also other parameters as interference is proposed, in the framework of partitioned systems. Also, we are considering include in future MILP models the possibility of frequency scaling in order to reduce power consumption while being in idle states. On the other hand, the exploration of Artificial Intelligence techniques for such tasks is to be considered, since it may be able to find a way to balance this reductions.

### Funding
This work was supported by Grant PID2021-124502OB-C41 funded by MCIN/AEI/ 10.13039/501100011033 and, by "ERDF A way of making Europe", by the "European Union". This work was also supported by PAID-10-20 (UPV). There was no additional external funding received for this study. The funders had no role in study design, data collection and analysis, decision to publish, or preparation of the manuscript.

### Grant Disclosures
The following grant information was disclosed by the authors:
PID2021-124502OB-C41.

MCIN/AEI/10.13039/501100011033.

"ERDF A way of making Europe".

"European Union".

PAID-10-20 (UPV).

## Competing Interests

The authors declare that they have no competing interests.

## Author Contributions

- Luis Ortiz conceived and designed the experiments, performed the experiments, analyzed the data, performed the computation work, prepared figures and/or tables, authored or reviewed drafts of the article, and approved the final draft.
- Ana Guasque conceived and designed the experiments, performed the experiments, analyzed the data, performed the computation work, prepared figures and/or tables, authored or reviewed drafts of the article, and approved the final draft.
- Patricia Balbastre conceived and designed the experiments, authored or reviewed drafts of the article, and approved the final draft.
- José Simó conceived and designed the experiments, authored or reviewed drafts of the article, and approved the final draft.

## Data Availability

The code is available at Gitlab and Zenodo:

- https://git.upv.es/gii-ai2-open/allocationpeerj.git.

- Ortiz, L., Guasque, A., Balbastre, P., & Simo, J. (2024). Allocation algorithms for multicore partitioned mixed-criticality real-time systems. Zenodo. https://doi.org/10.5281/zenodo.12731185.

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
