# Peer review of "Allocation algorithms for multicore partitioned mixed-criticality real-time systems"

_PeerJ Computer Science, doi:10.7717/peerj-cs.2609_

## Round 0.1 · original submission · Major Revisions

Based on the reviewers' comments, a "Major revision" will be required for this paper. Authors should enhance the explanation of the methodologies, interpretation of the results, and evaluation and verification of the works.

Reviewer 1 ·

Basic reporting

Some sections of the text may be confusing for readers. For instance, on lines 36-40, the authors mention the possibility of re-certification, but the explanation of these certifications isn't provided until the following paragraph (lines 41-51). Consider reordering the information for a smoother flow. Similar issues exist elsewhere, such as the explanation of line 176–177 formula's parameters being presented in a different order than the formula itself.

The introduction heavily emphasizes the hypervisor's importance, but this emphasis seems to diminish throughout the rest of the paper.

The font size used in the figures is difficult to read.

Figure 4 is not explained within the text.

I appreciate the well-referenced bibliography, related work, and the good explanation about the idea. I would also appreciate it if the code were made public.

Experimental design

The evaluation of the paper is a bit weak. The figures are not clear and not explained. For example, what is the meaning of each axis (Fig.5 shows different scenarios of number that are not explained in the paper), and a more in depth analysis will be appreciated.

It is not clear to which bar represent the proposal (WF or G), what is the baseline. Also, I would like to see a comparison against more allocation algorithms that are described in the related work.

In summary, I miss three points in the evaluation:
1) Including a comparison of the proposed idea against a wider range of allocation algorithms pointed out in the related work.
2) The descriptions of the figures need more details, like what each axis represents, what each bar represent.
3) The evaluation insights are not currently well-explained in detail.

Validity of the findings

The idea is interesting and, the results seem promising. The explanation of the idea is well done, with examples and easy to follow.

Reviewer 2 ·

Basic reporting

The paper considers partition/task allocation of mixed-criticality tasks to minimize the number of partition context switches and improve schedulability. This is achieved by partitioned scheduling on multicore platforms. This paper considers a single WCET for each task rather than using the common vestal task model, proposes two MILP formulations and a modified WFDU algorithm to map tasks into partitions/cores and uses an enumerative algorithm to search for the solution which provides the minimum of partition context switches.
A set of experiences was done and the results were provided.


(+)
+ The problem considered in this paper is an interesting problem
+ Globally, the paper is well written, and there are few typos
+ A rich and detailed state of the art section
+ An example is provided which allows you to better understand the methods
+ The organization of the paper is clear and easy to follow.



(-)

- The task allocation is a classical problem, the two proposed ILP formulations are basic
- Some basic results regarding the complexity of the problem (mostly NP-hard for each of the two phases) is not investigated, while the use of MILPs followed by a heuristic to find the solution which provides the minimum of partition context switches should be justified by these results.

- The method itself is very complex, requiring these phases.
- An exhaustive search is provided in Algorithm 1, for each allocation a scheduling plan is generated
* The complexity of the algorithm is not discussed and we have no idea how long it will take to find a solution.
* It is not clear how the number of context switches is computed?

- Some choices in the experimental setup should be revised.
- The use of two (one optimal, the other heuristic) phases is not justified. For example, for a similar (but not mixed-criticality)) problem, [A,B] are proposing a branch & bound addressing both placement and scheduling.




I believe that the paper needs to be reworked, especially regarding the second phase of the methodology.
The experimental section should be enhanced by giving results in terms of runtime. You can also propose an alternative to the enumerative algorithm which is exponential in terms of complexity.

Experimental design

no comment

Validity of the findings

no comment

Additional comments

(Comments)

- In the results part, you mentioned "the actual interference is measured ", how was it done ?!
- Were their any certification process used for this work?!
- Equations 3 and 4 could be gathered (page 5), the same for equations 11 and 12 (page 6)
- If the first task in a partition is allocated to a given core, then it follows that all the other tasks in that partition are allocated to the same core! Please be clear on this point
- Equation 14 will never hold due to the second part of the equation (j= P_(tau_0))?

(editorial comments)
- line 32: " .. but ... but ..", to be reformulated !
- line 45: " .. develop .."
- line 57: to be rephrased
- line 85: " it is "
- lines 99 -- 101: repetition
- line 132: in this line you use the acronym for the first time (EDF-VD)
- lines 144-146 : give citation


References:
[A] Peng, D. T., Shin, K. G., & Abdelzaher, T. F. (1997). Assignment and scheduling communicating periodic tasks in distributed real-time systems. IEEE Transactions on Software Engineering, 23(12), 745-758.
[B] Senoussaoui, I., Zahaf, H. E., Benhaoua, M. K., Lipari, G., & Olejnik, R. (2020, June). Allocation of real-time tasks onto identical core platforms under deferred fixed preemption-point model. In Proceedings of the 28th International Conference on Real-Time Networks and Systems (pp. 34-43).

Annotated reviews are not available for download in order to protect the identity of reviewers who chose to remain anonymous.

---

## Round 0.2 · Major Revisions

After the first revision, authors didn't put enough effort to address the comments from the reviewers. Hence, a "Major" revision is required. Authors should enhance the explanation of the methodologies, interpretation of the results, and evaluation and verification of the works as suggested by the reviewer.

Reviewer 1 ·

Basic reporting

All my concerns from my first review have been addressed. From my side, the article is ready to be published.

Experimental design

no comment

Validity of the findings

no comment

Reviewer 2 ·

Basic reporting

The authors relayed most of the questions and the reviews, Thank you! The paper is more clear, however, it needs to be completed, there are some points that need to be more detailed in the paper:

Concern#2: The two functions based on MILP are separated, I was wondering if there are some reasons why it is so important to make this choice, could we gather the two phases? Could we propose heuristics rather than MILP formulations? The authors should explain why it is important to separate the two problems and why they proceed to solve the general problem on two different phases.

Concern#4: Thank you for the results. The scenarios used in these experiences, including the number of tasks, their utilisation, and so on, should be provided by the authors.

Concern#5 : I can not find the explanation in the paper (neither at line 55 nor in the experimental section). I was wondering to know how actually you compute the number of context switches (Figure 6), it is important to know your method since this is the primary objective of this study.

Concern#7 : I will rephrase my question, why did you choose MILP-based formulations rather than other methods such as Brand and Bound based method. Your choice should be clearly justified.

Concern#10: Preemptive EDF scheduler is used in this paper. If two tasks are activated at different times, then, they do not interfere at run-time?! Please be clear on this point, if there are formulas or citations, then give them to explain properly, your idea.

Concern#13 : Thank you for the answer, please add this explanation to
the paper.


Other remarks :
— The schedulability section is poorly explained
— It is not clear what represent your x-axis in figure 5? You said, it represents each of the scenarios defined in Table 5, but actually it represents the scenario’s number, please be clear.
— How do you explain the fact that 'average(10C)' takes less time than 'average(8C)' in Figure 9 ? Does it continue to decrease ?
— Authors need to read the paper once again and correct miswriting.

Experimental design

no comment

Validity of the findings

no comment

Additional comments

no comment

---

## Round 0.3 · accepted · Accept

Authors have addressed all the comments from the reviewers. Hence, the paper is recommended to be published in its current form.

Reviewer 2 ·

Basic reporting

no comment

Experimental design

no comment

Validity of the findings

no comment

Additional comments

no comment